# Larval Dispersal Modeling Reveals Low Connectivity among National Marine Protected Areas in the Yellow and East China Seas

**DOI:** 10.3390/biology12030396

**Published:** 2023-03-02

**Authors:** Jiaying Lu, Yuanjie Chen, Zihan Wang, Feng Zhao, Yisen Zhong, Cong Zeng, Ling Cao

**Affiliations:** School of Oceanography, Shanghai Jiao Tong University, Shanghai 200240, China

**Keywords:** marine protected areas, connectivity, larval dispersal, biophysical modeling, network

## Abstract

**Simple Summary:**

In this study, a biophysical model was developed to assess the ecological connectivity of national marine protected areas in the Yellow and East China Seas. The results showed that ocean dynamics, pelagic larval duration, and distribution patterns had significant effects on larval dispersal. The existing national marine reserves in the Yellow and East China Seas did not form a well-connected network, and nearly 30% of them were isolated. The only connections were mostly from north to south. Four marine protected areas (all in coastal Zhejiang) emerged as key nodes for ensuring multi-generational connectivity. Due to the selection of study species with weak to strong potential dispersal, the results of this study can be applied to other organisms with similar life history characteristics, and can provide scientific reference for future reserve planning in coastal China.

**Abstract:**

Marine protected areas (MPAs) are vital for protecting biodiversity, maintaining ecosystem integrity, and tackling future climate change. The effectiveness of MPA networks relies on connectivity, yet connectivity assessments are often skipped in the planning process. Here we employed a multi-species biophysical model to examine the connectivity patterns formed among the 21 national MPAs in the Yellow and East China Seas. We simulated the potential larval dispersal of 14 oviparous species of five classes. Larvae of non-migratory species with pelagic larval duration (PLD) were assumed to be passive floating particles with no explicit vertical migration. A total of 217,000 particles were released according to spawning period, living depth, and species distribution, and they were assumed to move with currents during the PLD. Most larvae were dispersed around the MPAs (0–60 m isobaths) and consistent with the currents. Larval export increased with PLD and current velocity, but if PLD was too long, few larvae survived due to high daily mortality during pelagic dispersal. The overall connectivity pattern exhibited a north-to-south dispersal trend corresponding to coastal currents. Our results indicated that the national MPAs in the Yellow and East China Seas did not form a well-connected network and nearly 30% of them were isolated. These MPAs formed three distinct groups, one in the Yellow Sea ecoregion and two in the East China Sea ecoregion. Four MPAs (all in coastal Zhejiang) emerged as key nodes for ensuring multi-generational connectivity. Under the pressure of future climate change, high self-recruitment and low connectivity present significant challenges for building well-connected MPA networks. We suggest adding new protected areas as stepping stones for bioecological corridors. Focused protection of the Yellow Sea ecoregion could have a good effect on the southern part of the population recruitment downstream. Conservation management should be adjusted according to the life cycles and distributions of vulnerable species, as well as seasonal changes in coastal currents. This study provides a scientific basis for improving ecological connectivity and conservation effectiveness of MPAs in the Yellow and East China Seas.

## 1. Introduction

Marine ecosystems are the entities on which thousands of marine creatures depend for survival and reproduction. Human activities and climate change have put increasing pressure on marine ecosystems, leading to a decrease in marine biodiversity [1]. Marine protected areas (MPAs) are frequently established and powerful tools for conserving species. However, global marine biodiversity has continued to deteriorate [2], indicating a great need for more efficient MPA management. Building and improving ecological connectivity between MPAs can effectively assist in forming a well-connected MPA network [3]. An expanding body of empirical evidence has demonstrated the potential benefits of incorporating connectivity into conservation management [4,5]. Both the Aichi Biodiversity Target 11 (https://www.cbd.int/aichi-targets/target/11) (accessed on 3 August 2021) and the 2030 action target three of the Convention on Biological Diversity have called for ecological connectivity of MPAs as an essential criterion when evaluating the effectiveness of MPAs and accomplishing biodiversity goals [6].

The overall connectivity of a protected area network is reflected by studying the ecological connectivity of species in the protected areas, which is primarily affected by larval dispersal [7]. Many marine organisms have a pelagic larval phase and a relatively stationary adult phase, while the larval phase can be quite dispersive. Due to the fluidity and continuity of the ocean, protected species will move about and disperse between different habitats. The dispersal of larvae can greatly affect population dynamics [8]. Linkages between local populations are often maintained only through larval exchange between habitat patches. This is a fundamental ecological process that structures marine populations and confers ecosystems with resilience, and is thus important for planning MPAs [9,10]. Pelagic larval duration (PLD), often known as the time larvae spend as plankton and drift with currents, is the period between spawning and the juvenile stage of marine life history. In general, the longer the PLD, the greater the dispersal potential of the species, and the wider the distribution range; therefore, the larger the area to be protected [7].

Larval labeling [11,12], otolith microchemical analysis [13,14,15], genetic parentage methods [4,16,17], and biophysical modeling [18,19] are popular methods for assessing ecological connectivity in MPAs by studying larval dispersal. In the context of smaller spatiotemporal scales, larval labels, genetic parental analysis, otolith microchemical analysis, and landscape analysis are efficient techniques. However, these methods require intensive sampling and are usually expensive. Biophysical models can avoid such limitations and allow for accurate descriptions of larval dispersal on spatial and temporal scales [7,8,20,21,22,23]. At the beginning of MPA design, other methods are usually not applicable due to data and cost constraints. Biophysical models are therefore the preferred method for assessing the ecological connectivity of MPAs, especially at larger temporal and spatial scales [24]. The effects of larval behavior and topography on larval dispersal along the coast as well as population recruitment of the larval stages of species and connectivity between MPAs have been demonstrated by a number of realistic numerical models of coastal oceans [10,23,25,26]. These are ecologically pivotal phases of the dispersal process, but they are all correlated to some degree by the interactions of the characteristics of larvae (e.g., PLD) and the velocity fields of a particular ocean region.

MPAs have been in place in China for nearly six decades. As of 2021, 273 MPAs had been established, although the effectiveness of China’s MPAs has been impeded by accumulated issues over the years [27]. Most existing marine reserves were designed and managed according to limited local systems, and as such only few MPAs have involved ecological connectivity in marine conservation [28]. The connection patterns of existing protected areas are still unclear. Although connectivity was not considered in the initial construction of the reserves, it is likely that these MPAs are connected due to the movement of planktonic larvae under the influence of strong coastal currents. Due to the rich biodiversity and marine resources, MPA establishment in the Yellow and East China Seas has made significant progress. The Yellow and East China Seas initially formed a national MPA system with marine biodiversity protection as the core goal [29]. National MPAs have the highest protection level and represent the highest conservation value for biodiversity, but there are few studies on the connectivity assessment among these MPAs.

In this study, we assessed the ecological connectivity among the national MPAs of the Yellow and East China Seas, and examined whether the existing reserves have formed a well-connected network. By simulating potential larval dispersal patterns of species with different life histories and distribution characteristics, we investigated the importance of individual MPAs in strengthening ecological connectivity. This study incorporated multi-species and ecological connectivity into MPA planning, aiming to provide a scientific basis for improving the effectiveness of marine biodiversity conservation.

## 2. Materials and Methods

### 2.1. Study Region

Based on the consideration of comparing the connectivity between the Yellow Sea ecoregion and the East China Sea ecoregion, this study primarily focused on the continental shelf waters of the Yellow and East China Seas. The study area ranged from Lianyungang, Jiangsu Province, in the north to Dongshan County, Fujian Province, in the south (Figure 1). The study focused on 21 national MPAs (accounting for 27.3% of total national MPAs in China) and adjacent waters (24° N–36° N), involving fourteen special marine protected areas (SMPAs) and seven marine nature reserves (MNRs). MNR, in which extractive activities are highly restricted, and SMPA, including marine parks, in which multiple resource use is allowed. The ocean currents in the Yellow and East China Seas are from coastal currents (the Yellow Sea Coastal Current, the Yangtze Diluted Water, the Zhejiang–Fujian Coastal Current, and the Taiwan Warm Current) and the Kuroshio Current and its branches.

### 2.2. Ocean Model Configuration

Larval dispersal between MPAs is strongly influenced by ocean currents and is also a function of species’ life history [22]. For this reason, this study developed a biophysical modeling approach based on Regional Ocean Modeling System (ROMS) to quantify potential larval dispersal between MPAs in the Yellow and East China Seas. ROMS is a free-surface, terrain-following, three-dimensional nonlinear baroclinic ocean model widely used for larval dispersal in connectivity modeling [23,34,35]. This model solves the Reynolds-Averaged Navier–Stokes equations based on hydrostatics and Boussinesq approximation in sigma terrain-following coordinates (S-levels) and a curvilinear orthonormal Arakawa C grid over the vertical and horizontal axes, respectively [35]. The modeled area of this study followed Chen (2020) [31] and was located at 117° E–135° E, 24° N–42° N, including the Bohai Sea, the Yellow Sea, the East China Sea, the Korean Strait, part of the Sea of Japan, and the Northwest Pacific Ocean [31]. The model grid mesh spanned 730 × 438 cells in the horizontal direction (Appendix A) and contained 20 vertical layers. Horizontal grid density was enhanced near the coast of mainland China. The horizontal resolution varied from ~1 km near the Yangtze Estuary to ~6 km near the east open boundary.

The bathymetric data came from two sources: (1) digital chart data with high accuracy were used in the Yangtze Estuary, Hangzhou Bay, and Subei Shoal; (2) ETOPO1 data with a resolution of 1/60° were used in other sea areas. Since this study focused on the continental shelf area, the minimum depth of water was taken as 5 m. The maximum depth was 2000 m. The model northern and western boundaries were closed, and the eastern and southern boundaries were open boundaries employing the Chapman scheme [36] and the Flather scheme [37] for the free-surface and two-dimensional velocities, respectively. The atmospheric forcing and boundary forcing data were acquired from the ECMWF (The European Centre for Medium-Range Weather Forecasts) ERA-Interim reanalysis product and the Copernicus global ocean analysis product. The tidal harmonic constants on the open boundary were taken from the Global Tidal Database of Oregon State University (http://volkov.oce.orst.edu/tides/) (accessed on 7 March 2019) comprising 13 constituents (M2, S2, N2, K2, K1, O1, P1, Q1, M4, MS4, MN4, MM, and MF). The time step for the inner mode was 120 s, and the time step for the outer mode was 6 s. The model was continuously integrated from the initial stationary state. The simulation time ranged from 1 January to 31 December 2016, because the particle dispersal dynamic in 2016 was close to the average field in recent eight years (see Appendix A). The verification results showed that the model could simulate the dynamic characteristics of the currents in the Yellow and East China Seas [31].

### 2.3. Lagrangian Particle Dispersion Model

A Lagrangian particle dispersion model-based ROMS was used to simulate larval dispersal of different species. In the model, particles that moved passively with ocean currents represented the larvae of species. These particles were released at specific locations, times, and water depths according to their life history characteristics, regardless of active swimming and vertical migration. To analyze the ecological connectivity of different species in the Yellow and East China Sea national marine reserves, this study focused on marine organisms distributed in the abovementioned areas with high conservation or fishery value. Representative species for connectivity assessment were selected from (1) the species assessed as critically endangered (CR), endangered (EN), or vulnerable (VU) in the IUCN Red List of Threatened Species, and (2) the main economic species in The Atlas of Main economic species in the East China Sea. A species could be selected if it met the criteria of (1) living in one of the 21 MPAs in the Yellow and East China Seas, (2) belonging to marine groups, (3) having a planktonic larval stage, and (4) having exhaustive life history information.

The biological parameters of the model included life history, distribution information, pelagic larval duration (PLD), living depth, and spawning window [38] that have been demonstrated to play important roles in ecological connectivity. The data were collected from published articles and public databases (Appendix A). To make the research more general, study species were selected to represent other organisms with similar life history characteristics (pelagic larval duration, living depth, and spawning season) and covering as diverse a range of each biological parameter as possible. The process was completed on 8 June 2022, resulting in 14 species (Table 1) in five classes (Actinopterygii, Actinozoa, Crustacea, Holothuroidea, Cephalopoda). These species represented a range of dispersal phenotypes (from shorter to longer dispersal distances). The PLD of the 14 species ranged from 7 to 75 days and was set as the floats’ release duration in the model. Timing of spawning was defined as the time when spawning events happened during the entire year.

All of these species only had one spawning window per year. The particles were released at 12:00 a.m. in the middle day of the spawning window. The particles were released according to the distribution of species [39]. Once a species was likely to occur in any MPA, the central point (with a resolution of 0.0001°) of the MPA was set as the starting point. For one species, 1000 particles were released from each starting point. Passive particles with a constant size were released without vertical migration, ontogenetic changes, or swimming capabilities between the depth of 5 and 100 m in terms of their habitats. The position of particles was output every six hours. Both central points of MPAs and the number of particles released for each species are displayed in Table 2. For example, for *Anguilla japonica* that occurred in 21 MPAs, 21,000 particles were released from MPAs and floated passively for 10 days. Although the influence of mortality on connectivity was widely recognized [7], we had little information about the exact effects (i.e., sea temperature) on larval mortality since the field-based data are limited. The mortality was usually set at a fixed value in the studies of MPA connectivity [23,40]. It is known that the first stages of larval life are characterized by a high daily mortality rate that ranges from 10% to 20% [40,41]. Thus, this study included mortality rates, and 15% of the particles of each species were removed daily and at random to obtain effective connectivity information during model output analysis.

Based on the data obtained from the abovementioned models, species dispersal trajectories were drawn to determine whether the established MPA network could cover the species’ corridors. The dispersal fluxes of 14 species to each protected area were calculated to represent the accessibility of species to the protected area (taking the actual boundary of MPAs as the range). The dispersal flux d(*i*) was defined as the particle abundance of each species at 21 national MPAs. Dispersal fluxes of some species in MPA *i* were calculated by the fraction of the cumulative quantity of larvae particles entering MPA *i*, r(*i*) to the total quantity of particles released from all k starting points.
(1)$$d\left(i\right)=r\left(i\right)/\underset{x}{\sum }r\left(i\right)$$

However, higher larval abundance in one national MPA could only indicate that the accessibility of the area for that species was better, and this would not explain population recruitment of species and connection probability c(*i*,*j*) between MPAs. Therefore, further analysis of self-recruitment rate selfr(*i*) and subsidy recruitment rate subr(*i*,*j*) of the protected area needed to be included. Here, each protected area was regarded as a node in the network. The connection probability was the proportion of larvae departing from MPA *i* that arrived in MPA *j* [42]. The self-recruitment rate was defined as the proportion of larvae that originated from a MPA and reached the same MPA. The subsidy recruitment rate of a single MPA was the proportion of larval replenishment coming from other MPAs summed over all MPAs and species [8]. The connectivity probability revealed the relative contribution of every MPA to surrounding protected areas in terms of recruitment, i.e., sources and sinks of the population recruitment. Protected areas with high source intensity could promote biodiversity conservation of both protected species and economically important species, and thus they were crucial for the entire MPA network [43].
(2)$$\mathrm{selfr}\left(i\right)=c\left(i,i\right)/\underset{j}{\sum }c\left(i,j\right)$$
(3)$$\mathrm{subr}\left(i,j\right)=\underset{j\neq i}{\sum }c\left(i,j\right)/\underset{j}{\sum }c\left(i,j\right)$$

Patterns of larval abundance at the end of the pelagic larval duration demonstrated the potential larval dispersal. Since MPAs’ size, location, time since establishment, fishing restrictions, and regulation enforcement are important factors affecting larval dispersal [44,45], larval export was likely to vary among MPAs. Because of the lack of data, this study only involved MPA size, and the MPAs in the pattern were drawn with weights in terms of their sizes.

Graph theory was used to quantify each national MPA’s ecological connectivity within the network [8,43]. Graph theory is widely used to measure and visualize connectivity patterns between reserves in a lot of related research [7,8,20,21,23]. Nodes and edges of the graph were defined by MPAs and larval trajectories, respectively [46]. Two network node importance metrics were illustrated: (1) Degree centrality of node *i* was the number of nodes connected to node *i*. The assumption was that the crucial MPAs were those that had many connections. The more connections an MPA had, the stronger its influence. Since the connection between two protected areas was directional, we used indegree and outdegree of graph theory to quantitatively analyze the role of each MPA in larval recruitment. They referred to the set of MPAs connected immediately upstream (Indegree of nodes) and downstream (Outdegree of nodes). (2) Betweenness centrality of node *i* was the sum of all the shortest paths connecting nodes x and y across node *i,* according to Andrello et al. [8] and Treml et al. [43]:(4)$$bc\left(i\right)=\underset{x\neq y\neq i}{\sum }\frac{\sigma_{xy}\left(i\right)}{\sigma_{xy}}\;$$
where *σ_xy_* is the total number of shortest paths between node x and node y, and *σ_xy_ (i)* is the number of those passing across the node *i*. When incorporating connectivity into the design of an MPA network, analysis of the effect that every MPA plays in promoting connectivity has been demonstrated to be significant [47]. As an MPA is connected with other MPAs in a multi-step way, each act as an essential node to transfer genes and individuals between MPAs that are not directly connected. The roles of such central MPAs were identified by betweenness centrality [48]. As a result, betweenness centrality demonstrates the significance of each node in acting as a corridor for propagating larvae across the network [46,48] and is related to the landscape ecology concept of “stepping stones” [42]. Degree centrality and betweenness centrality are therefore two complementary node-level metrics that quantify the connectivity of single MPAs in relation to their closest neighbors (degree centrality) or to all nodes within the network (betweenness centrality) [8]. The network connectivity variables were obtained by the equal-proportion addition of threatened species and economically important species. All metrics were calculated by the igraph package in R [49].

## 3. Results

### 3.1. Larval Dispersal

The entire flow field simulated by the model (Figure 2) was consistent with the results of previous studies [30,31]. The velocity of most of the study areas was 0.1–0.2 m/s. The dominant current of the Yellow Sea ecoregion (i.e., YSCC) was much weaker than that of the East China Sea ecoregion. Outside the mouth of the Yangtze River, the YDW generally flows to the northeast during the southerly winds in Seasons 2 and 3. In Season 1, the runoff of the Yangtze River decreases greatly, and under the northerly winds the YDW flows southward along the coast. ZFCC was dominant along the coast of Zhejiang and Fujian provinces in the East China Sea ecoregion. Because this flow was located close to the MPAs, it may be the ocean current that has the greatest impact on population connectivity between protected areas in the study region. The upstream area of ZFCC (near the Yangtze River and the Hangzhou Bay) had the highest velocity, especially during Season 4 (October–February) due to the stable and strong northeast winds. The TWC is the current occurring on the east side of the East China Sea coastal current and south of the Yangtze Estuary. This current flowed to the northeast along the coast of Fujian and Zhejiang almost all year round, except in Season 4 when the surface layer was easily affected by the northerly monsoon and the flow direction turned southward.

Most of the particles were scattered within the 40 m isobath and radiated outward from the released points. As a result of a combination of biological characteristics, ocean currents and habitat distribution, the potential dispersal path of different species varied widely (Appendix A). Similar to previous studies, for most species, the longer the planktonic larval stage, the wider the larval dispersal range, and the higher the degree of connectivity between protected areas [21]. However, when the PLD was too long, for example, in *Nemipterus virgatus* with the longest PLD (75 days), due to high daily mortality during the floating period the survival rate of larvae decreased rapidly, and thus the ecological connectivity of this specie was limited in the study area (Appendix A).

Ocean current directions and the continental shelf appeared to account for the spatial patterns of larval dispersal (Figure 3). Larvae could accumulate even though they floated far from their original location (the Yangtze River estuary), where the continental shelf was large (40–60 m isobath). The directions of ocean currents dominated larval transport. Currents were likely to carry larvae for long distances to areas unsuitable for settlement. Because the velocity of surface water was greater than in deeper layers, species living in shallow water were more likely to reach distant MPAs. Although the larval concentration of different species differed widely in the pelagic part, similar aggregation phenomena existed in the nearshore reserve. The floating trajectory of the particles extended as far as the 60 m isobath sea area, and the diffusion path was highly consistent with the direction and intensity of ocean currents in four seasons.

Since the national MPAs were usually close to the coast, larval dispersal was primarily affected by coastal currents and the Kuroshio branches. Under the influence of the Yellow Sea Coastal Current and the Taiwan Warm Current as well as the Kuroshio current branches, the particles from the north and south gathered together in the Yangtze River estuary with high abundance. The current velocity near the Yangtze River was relatively high. Because of the Yangtze-diluted water, particles had a clear trajectory from west to east and moved northward from winter to summer. Due to the variation in the direction and velocity of currents in the four seasons, the dispersion patterns of particles released in different spawning seasons showed significant differences. Spawning from October to December may be conducive to larval diffusion and enhanced species communication. Larval export distance increased between Season 1 and Season 2 owing to the strengthening of coastal current velocity (from south to north). Because of the fast flow at the Yangtze River estuary from April to June, even though the PLD of *Anguilla japonica* was short, there was still pronounced particle aggregation near the ocean currents.

The number of distribution points affected larval dispersal in this study. Due to the wide distribution of economically important species, their dispersal fluxes in 21 MPAs were larger than those of endangered species, especially *Penaeus japonicus* and *Anguilla japonica* (Figure 4). Conversely, since *Acropora solitaryensis* only occurred at two MPAs according to the assessment of suitable habitats (Table 2), the distribution characteristic resulted in high concentration in a single area. High larval abundance was found in JS-2, JS-5, ZJ-3, ZJ-9, FJ-3, and FJ-5, indicating that the MPAs mentioned above were focal reserves with higher species diversity. Despite the distribution points, species with short PLD had higher abundance in the MPAs (e.g., *Acropora solitaryensis, Epinephelus akaara, Penaeus japonicus*, and *Anguilla japonica*). When the PLD was ≥ 45 days (the top three rows in Figure 4), larvae followed the currents for longer periods of time, floating far beyond the MPA boundary limits relative to their size, which reduced the proportions remaining in protected areas. However, the high abundance of particles in a protected area could only indicate that the MPA was highly accessible for certain species, and this could not explain the population recruitment of species and connection among 21 MPAs in the network. Therefore, recruitment rate calculation and directed network analysis were required.

### 3.2. MPA Ecological Connectivity Analysis

Under comprehensive consideration of different types of representative species in the Yellow and East China Seas, the established MPAs did not form a connected protected area network. The self-recruitment fraction of all protected areas was far higher than the subsidized recruitment fraction and was highly variable among MPAs (Appendix A). Self-recruitment fractions were distributed serially in the range of 0 and 1 (median 0.029, interquartile range 0.0415). Six MPAs relied totally on self-recruitment (JS-1, JS-4, JS-5, SH-1, SH-2, FJ-5), while seven MPAs had zero self-recruitment (JS-3, ZJ-1, ZJ-2, ZJ-4, ZJ-7, ZJ-8, FJ-4). Total self-recruitment indicated that the majority of larvae tended to stay in the original reserves rather than migrate to adjacent protected areas that showed zero self-recruitment. MPAs ZJ-1, ZJ-2, and ZJ-8 relied only on subsidized recruitment. As a consequence, the ecological connectivity of MPAs in the Yellow and East China Seas was very low. MPA FJ-5 had the highest self-recruitment fraction (0.08). The highest subsidized recruitment fraction was from MPA ZJ-8 to ZJ-9 (0.02). The high dispersal fluxes shown in Figure 4 of MPA JS-5, ZJ-3, FJ-3, and FJ-5 were due to the larvae distributed in these MPAs not leaving the MPAs or returning to the original MPAs, resulting in high rates of self-recruitment.

Consistent with the results in Appendix A, the subsidized recruitment fraction between connected MPAs (Figure 5) was always very low, mostly in the range of 0.0001–0.001 (median 0.00019, interquartile range 0.00077). Among the Yellow Sea ecoregion, JS-3 contributed high larval supplementation to JS-2, where the connection direction was from south to north. No connection was observed between other protected areas in the region. The population replenishment between MPAs located at the coast of Zhejiang in the East China Sea ecoregion was clearly better than those of other provinces. Most larvae were exported from northern protected areas to southern protected areas, and only four connections were from south to north. Among the latter, MPA ZJ-2 to ZJ-6 and ZJ-8 to ZJ-3 and ZJ-9 had the highest recruitment. As to Fujian province in the south of the East China Sea ecoregion, MPA FJ-4 had larvae supplemented to FJ-2 and FJ-3 movement from south to north.

The degree of centrality (indegree and outdegree) and betweenness centrality varied greatly across MPAs (Figure 5), indicating the importance of individual MPAs. The degree pattern clearly showed that MPAs along Zhejiang province had a higher degree (Figure 5d) and betweenness centrality (Figure 5a) among the 21 MPAs, probably due to their central positions where the estuary of the Yangtze River met the coastal currents. MPA ZJ-2, ZJ-3, and ZJ-5 had the highest degree (which was four), making the most connections to other protected areas. Having the highest outdegree (which was three), MPA ZJ-2 was an important source for the surrounding reserves (Figure 5c). Meanwhile, the degree of MPA ZJ-5 was the same as the indegree (Figure 5d), i.e., its connections were all due to larval supplementation from other protected areas. In particular, due to the frequent southward connections, MPA ZJ-2 and ZJ-3 possessed high betweenness centrality. These MPAs thus were inferred as key nodes to ensure the connectivity between the Yellow Sea and East China Sea ecoregions. The highest betweenness centrality occurred in MPA ZJ-3 (which was ten), showing that it was the most critical MPA in the entire network.

To summarize, MPAs in the Yellow and East China Seas did not form a systematic protection network, as the connections between MPAs were always one-way only. There were three MPA clusters located in Jiangsu, Shanghai, and Zhejiang, and Fujian coastal areas. Jiangsu and Fujian MPA clusters only involved 2–3 MPAs, and all of the connections were from south to north. The ecological connectivity was very low, and MPAs were relatively isolated. The Shanghai and Zhejiang (SZ) MPA cluster covered at least nine MPAs that had larval export or recruitment with other MPAs (Figure 5a). Moreover, the MPAs of this cluster had the highest degree of centrality among all of the studied MPAs, and five MPAs of the cluster were demonstrated to be crucial “stepping stones” (high betweenness centrality), and thus ecological connectivity was highly strengthened by frequent multidirectional connections. In addition, MPA ZJ-9 seemed to be the “hub” of the Fujian MPA cluster and the SZ MPA cluster, as it was the closest geographically to MPA FJ-1.

## 4. Discussion

MPAs are widely identified as useful tools in biodiversity conservation and sustainable fisheries research, while their effectiveness relies on species larval export and ecological connectivity between MPAs [4,5,26]. Taking representative species with fishery and conservation value as a case study, this paper has revealed that (1) the connectivity between national MPAs in the Yellow and East China Seas was very low, and the system did not form a connected network; (2) larval dispersal and connectivity were primarily affected by coastal currents, PLD, and species occurrence; (3) individual MPAs can be crucial sources or sinks of populations (high recruitment) and can be essential in maintaining the connectivity of entire system (high degree and betweenness centrality).

### 4.1. Model Strengths and Limitations

Early studies on MPAs employed markers and sampling methods [12,14]. More recently, genetic analysis and microchemical methods have been significant developments and extended to deduce patterns of ecological connectivity and larval dispersal [15,16,26]. These methods incur high costs and involve high-intensity sampling and experiments for analysis, and thus they can barely be used on wider spatial and time scales. In this context, the advantages of biophysical modeling are highlighted due to their low cost, no sampling requirement, and applicability to broad spatial and temporal scales with the support of sufficient biological and hydrodynamic data. This study is the first evaluation of ecological connectivity among several MPAs at the scale of entire seas in China.

Although there is a trend toward increasing the number and coverage of China’s MPAs, biodiversity conservation is not working well [2]. One of the most mentioned problems is the decentralization due to the lack of systematic planning and design that demands a connectivity assessment of the whole system and the establishment of an effective MPA network in China [28]. Sustainable population recruitment determined by larval dispersal is more important for conservation diversity than MPA size and shape [20]. The model presented in this paper can be taken as one of the first steps in attempting to achieve this task by biophysical modeling.

Nonetheless, this study has four main limitations. First, although the horizontal grid size of ROMS was narrowed near the coast (the resolution near the Yangtze River estuary is precise to 1 km), larval export and connectivity can be affected by ocean processes at much smaller scales [50]. Models with a higher resolution can better simulate the demographics of larval dispersal [51].

Second, the larval export was predicted under the hypothesis that larvae were passive particles without growth or voluntary movement and behavior, while vertical migration, active movement, and interaction with resident species can change the patterns of larval dispersal [21,22]. The passive floating that only relied on currents may underestimate the probability of recruitment [38]. Future models that fully considered the abovementioned biological factors would be conducive to biological process research [52].

Third, due to the lack of latest data, the life cycle information of studied species was mostly from the literature and research from past decades (Appendix A). The biological data from recent years will be beneficial for the prediction results that would be closer to the actual larval export [7]. In addition, this study simulated the larval dispersal over one year. Considering the impact of climate change and other environmental variables on biological behaviors and hydrodynamic modes over decades, it is possible that there may be significant differences in the connectivity patterns from year to year [53].

Finally, species can also spawn and reproduce in unprotected areas or in other MPAs with lower conservation levels. The individuals from these spawning and nursery grounds can also recruit the national MPAs involved in this study and strengthen the larval abundance. From this perspective, the connectivity may be underestimated. Moreover, subsidized recruitment depends on the productivity gap between protected areas and fished areas [23]. Potential larval supply from outside the MPAs is unknown due to the lack of productivity data both within and outside the MPAs. Thus, predicting whether the number of larvae exported is adequate to lead to recruitment supply remains challenging.

In consideration of these restrictions, the present results should be taken with caution. A model with a higher resolution, a better understanding of larval biology, and finer data for productivity, biomass, and intensity of fishery could optimize the analyses and results in this work [51]. This study conducted larval dispersal simulations on species with different dispersal strategies and analyzed the effects of ocean currents, pelagic larval duration, living depth, and spawning month on larval dispersal, taking into account survival rates. Instead of assuming that all species occur in each MPA, simulation for larvae was carried out based on previous suitable habitat studies on related species, with the goal to approximate actual larval dispersal. In spite of all these restrictions, the results remained conservative, as this work considered an optimistic scenario when assessing connectivity. That is, fisheries were effectively confined in all MPAs in the Yellow and East China Seas, and the moderate larval pelagic mortality was taken in the analyses.

### 4.2. Ecological Connectivity between the MPAs

The main result of this study is that the Yellow and East China Sea MPAs did not form a well-connected network, as there was very low ecological connectivity. Larvae spread to a limited number of adjacent MPAs, and most did not settle in the protected areas. The number of connections generated by 14 species between 21 MPAs was only 33 (degree). In addition, 88% of connections were from northern to southern MPAs, and MPAs in Jiangsu and Fujian only had northerly larval export. Six MPAs were totally isolated, and 70% of the MPAs had connections, while most of these were unidirectional, highlighting the weak connectivity. Moreover, the connectivity in the East China Sea was much better than in the Yellow Sea ecoregion. The notable finding was that Zhejiang MPAs had a relatively tight cluster involving important sinks and sources (indegree and outdegree) compared with the coasts of other provinces. Apart from the small number of connections between MPAs, the connection probabilities were always low. The low connection probability (<0.001) indicated that the population migration caused by larval dispersal was insufficient to influence demography [53] but may be sufficient for genetic differentiation and evolution [8]. In this work, only 30% of the connection probabilities between MPAs in the Yellow and East China Seas were above this value, meaning that these connections can be considered demographically relevant.

Most connections between MPAs were consistent with the direction of major coastal currents in the Yellow and East China Seas. In spite of this, because the distribution data in this study were derived from the results of suitable habitat model simulation, the consequences of larval dispersal and recruitment were supposed to be interpreted as potential settlement availability. Since the three major coastal currents (YSCC, YDW, ZFCC) possibly change their directions during different seasons, the larval export of species with distinct spawning times showed various patterns, especially in winter and summer. The MPAs along the coasts of Zhejiang and Fujian provinces showed frequent bidirectional connections, although the southern direction connections dominated (Figure 5). We attributed this to the fact that most of the studied species spawn from December to June, which are critical period for marine fish breeding in the Yellow and East China Seas. The dominant coastal current, ZFCC, is strong and southward in spring and winter but weak and northward in summer and autumn. During the spawning time, southerly currents are strong. As a result, most larvae flowed from the northern MPAs to the southern MPAs. Less directional connectivity from southern to northern MPAs was observed in Jiangsu and Fujian provinces. It seems the connections were determined by summer coastal currents that were probably due to the greater distribution of species spawning in Season 2 in the MPAs.

MPAs located upstream of the coastal currents (ZJ-3, ZJ-4, ZJ-5) are important larval sources for downstream reserves. Because these currents span long distances, after having received larvae from upstream MPAs, these MPAs could act as sources of larvae for the southern protected areas, ensuring sufficient connectivity in this zone. Larval abundance was generally high in the waters around the Yangtze Estuary that is likely to be an area with rich biodiversity due to the transport of sediments and nutrients by the currents [31]. This result highlights the need for the incorporation of these areas in the current MPA zonation to improve the MPA effectiveness. ZJ-9 could act as a bridge between the Zhejiang and Fujian coasts thanks to relatively high MPA density and the opposite flow of ZFCC in summer and winter, leading to a Zhejiang MPA cluster. Compared with Zhejiang MPAs, the low MPAs density of the other three provinces indicates that connectivity may depend on the number and spatial arrangement of MPAs. In other areas of the globe where MPAs are fewer, connectivity will be likely lower than in the East China Sea.

Populations of almost 50% of the MPAs were only replenished by locally bred larvae, and there was no supply from other MPAs. Previous studies have also found low connectivity and high self-recruitment using biophysical models for other seas [7,38]. Low connectivity and high self-recruitment may lead to detrimental results in the adjustment of nearby populations. A population that depends on self-recruitment can be demographically stable. However, even if new recruits are adequate, the lack of larval supply from other populations can result in inbreeding depression, increasing the risk of population extinction [54]. Moreover, since isolated populations cannot send new adaptive alleles to other populations, weak connectivity may decrease the probability of the population adapting to global climate change [55].

### 4.3. Future Planning of the MPA Network

China is currently implementing a strategy of “eco-civilization,” with MPAs expected to be one approach to achieving sustainable marine ecosystems [56]. More protected areas will be established along coastal China and some isolated nodes are likely to be integrated, which is very promising to form networks of MPAs. MNRs are designed to avoid disturbances by prohibiting human activities strictly, whereas SMPAs allow multiple use of marine resources with limited extractive activities [28]. Ideally, MNRs are preferred to maximize the connectivity, but SMPAs are more operational when considering social and economic development.

Larval dispersal to surrounding areas hinges on MPA areas and spacing between MPAs [57]. Expanding the area of these reserves and setting them closer together would be beneficial in enhancing the ecological connectivity of the southern Yellow and East China Seas. MPA ZJ-2, ZJ-3, and ZJ-5 played crucial roles as sources or sinks of recruits for surrounding MPAs as well as were key gateways in sustaining the connectivity of the entire network (Figure 5). Taking them as the focus of MPA designation can have spillover effects on the surrounding reserves that may effectively strengthen ecological connectivity and improve the effectiveness of marine biodiversity protection in the entire area. In addition, high self-recruitment MPAs should be given priority protection, for example, MPA JS-2, JS-5, FJ-3, and FJ-5. Increasing their protection level with stricter fishing bans can be effective to the conservation of biodiversity. Additional protected areas should be placed in areas identified as being crucial for genetic diversity conservation, paving stepping stones between existing MPAs and enhancing connectivity [25,58].

The need for customized and differentiated conservation strategies for different protected species should be emphasized [59]. The potential ecological corridors of 14 general species types that this study considered are the reference sites from which new protected areas can be established (Appendix A). For species with weak dispersal potential, strong sessile ability, and scattered distributions such as *Acropora solitaryensis* and *Apostichopus japonicus*, relatively small protected areas that cover the habitats they need would be reasonable under the condition of saving conservation costs. Vulnerable species such as corals, which are confined to a single protected area, usually live in the same area for their entire life, and thus enhanced habitat protection and management in the original area can be best. For species with strong dispersal potential and wide distributions such as *Epinephelus akaara*, *Anguilla japonica*, and *Larimichthys crocea*, these circulate in multiple MPAs during the planktonic larval stage, and thus it may be necessary to establish a large network of protected areas to form an ecological corridor.

The population replenishment was generally from the northern protected areas to the southern protected areas. Thus, new protected areas could be built at the ecological corridors along the Yellow Sea ecoregion (Jiangsu coast) first. Focused protection of the northern seas could lead to multi-step connections with protected areas in the south, promoting population connectivity and gene exchange between the north and the south. Moreover, the currents in the Yellow and East China Seas have strong seasonal differences, carrying larvae in different directions. Therefore, more protected areas should be established along the gateways according to the seasonal variation of connectivity and management plans need to be adjusted in different seasons to balance development and conservation. It has been reported that the Yellow and East China Seas have experienced warming at twice the global average since 2011 [60]. Local populations in the northern parts of the Yellow and East China Seas have possibly not adapted to the warmer water. The lack of population recruitment from south, warm-adapted populations can weaken the species sustainability in the north of Yellow and East China Seas. In climate change scenarios, genetic thermal stress resistance is likely to be the basis for species to adapt to ocean warming. These populations with higher resistance should be established in areas that can support more vulnerable communities [55]. Therefore, it is essential to research the effects of ocean warming and variation in current velocity and direction on larval floating and survival in future studies [61].

Increasing numbers of studies have begun to use multiple methods [62,63]. This paper revealed weak connections between the MPAs in the Yellow and East China Seas using oceanographic dynamics and biological information, while approaches that capitalize on the complementary strengths of genetic parentage datasets and biophysical models can produce accurate larval dispersal patterns at regional scales [64]. Connectivity among MPAs in the Yellow and East China Seas has hardly been the object of genetics studies. Thus, it is necessary that future research identifies the hereditary stability of species protected in MPAs to explain the geographic distribution of adaptive genetic variation.

## 5. Conclusions

We used a biophysical model to simulate larval dispersal in national reserves in the Yellow and East China Seas to assess connectivity between MPAs. Evaluating the larval recruitment of study region and larval supply outside MPAs is challenging due to the lack of information of larval biology, and the productivity and intensity of fishing in MPAs. Our study revealed a high self-recruitment rate and weak connectivity of the present MPAs system in the Yellow and East China Seas, demonstrating potentially deleterious consequences for conservation effectiveness and population persistence. The connectivity patterns are likely to be the results of ocean currents, species occurrence, and life cycle. The future MPA design needs to establish a functioning network with better connectivity within and between MPAs with the consideration of essential factors above, global climate change as well as economic and social costs. Further studies are needed to combine different methods to explore larval connectivity changes and their effects on the population recruitment within and outside MPAs.

## Figures and Tables

**Figure 1 biology-12-00396-f001:**
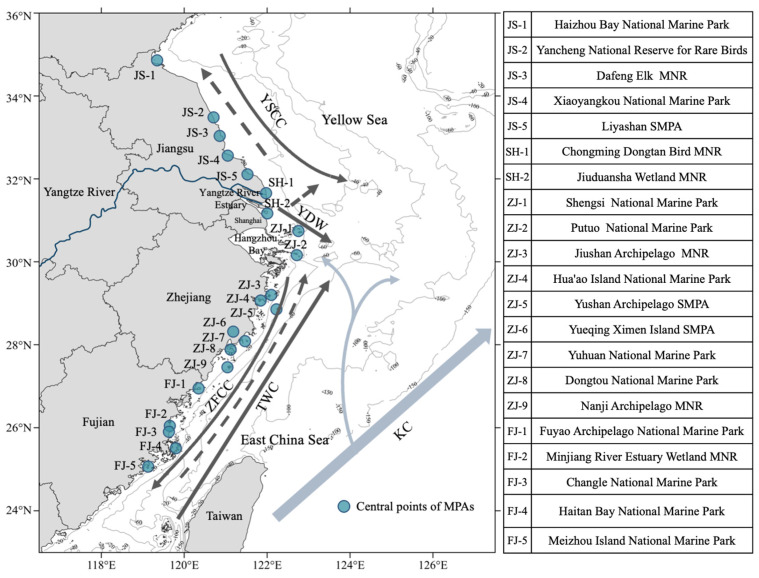
Distribution of 21 national MPAs and major currents in the Yellow and East China Seas. The blue points are the central points of MPAs. Arrowhead lines show the current directions. Dashed arrowhead lines are the possible directions of YSCC, YDW, and ZFCC in summer [30,31,32,33]. YSCC: the Yellow Sea Coastal Current; YDW: Yangtze Diluted Water; ZFCC: Zhejiang–Fujian Coastal Current; TWC: Taiwan Warm Current; KC: Kuroshio Current. MNR and SMPA in right table were marine nature reserves and special marine protected areas, respectively.

**Figure 2 biology-12-00396-f002:**
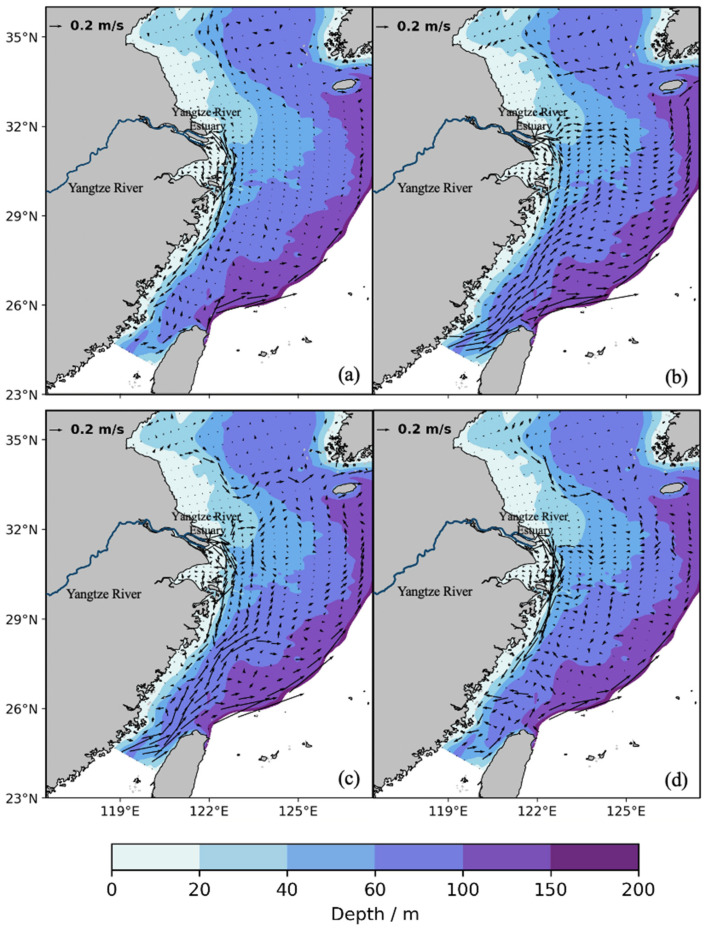
Modeling quarterly mean velocity and direction of ocean currents in Yellow and East China Seas: (**a**) January–March, (**b**) April–June, (**c**) July–September, and (**d**) October–December. The length of the line indicates the average velocity, and the arrows indicate the current direction.

**Figure 3 biology-12-00396-f003:**
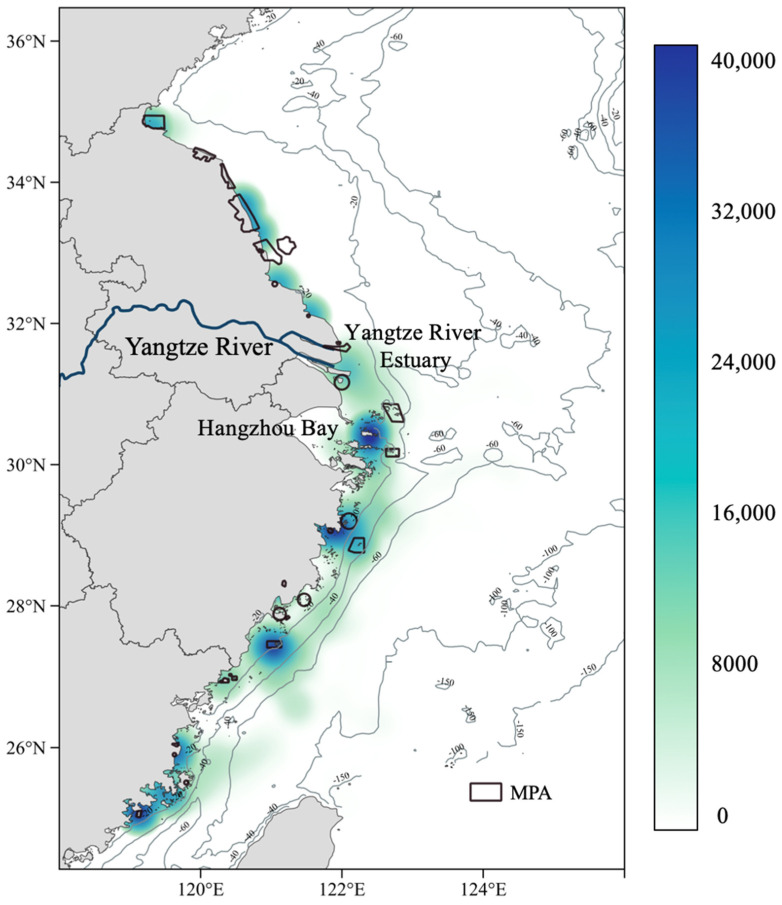
Total larval abundance at the end of pelagic larval duration. The color indicates the density particles. Particles of all 14 species were superimposed and were given the same weight.

**Figure 4 biology-12-00396-f004:**
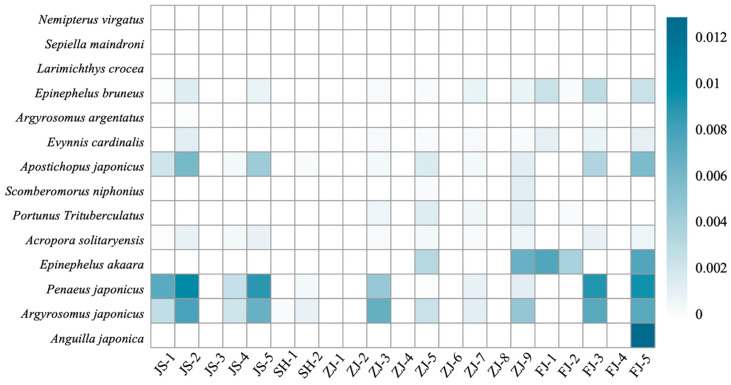
Dispersal fluxes of 14 species in each MPA. The species are arranged from top to bottom according to the length of PLD.

**Figure 5 biology-12-00396-f005:**
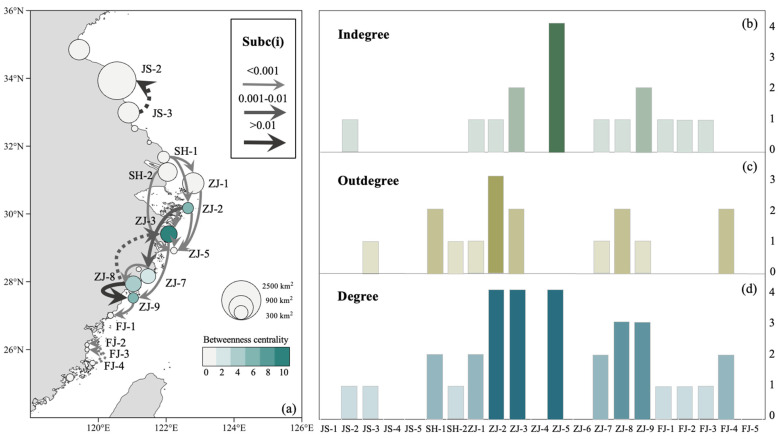
Schematic diagram of connections between MPAs and the network node importance metrics. Circle sizes indicate the relative sizes of protected areas. The directions of the arrows represent the direction of species particle dispersal. Dashed lines indicate from south to north, and solid lines indicate from north to south. Thickness of the lines represents the subsidized recruitment rate between MPAs. The network node importance metrics are shown in (**a**) Betweenness centrality, (**b**) Indegree centrality, (**c**) Outdegree centrality, and (**d**) Degree centrality.

**Table 1 biology-12-00396-t001:** The biological parameters of species included in the model.

Scientific Name	Classification	Category	PLD (Days)	Depth (m)	Spawning Window
*Nemipterus virgatus*	Actinopterygii	Threatened	75	25	April–May
*Sepiella maindroni*	Cephalopoda	Economic	45	10	April–May
*Epinephelus bruneus*	Actinopterygii	Threatened	45	20	May–June
*Larimichthys crocea*	Actinopterygii	Threatened	33	15	April–May
*Argyrosomus argentatus*	Actinopterygii	Economic	33	50	June–August
*Argyrosomus japonicus*	Actinopterygii	Threatened	33	100	January–March
*Evynnis cardinalis*	Actinopterygii	Threatened	30	45	November–January
*Apostichopus japonicus*	Holothuroidea	Threatened	20	10	May–June
*Scomberomorus niphonius*	Actinopterygii	Economic	19	20	May–June
*Portunus Trituberculatus*	Malacostraca	Economic	17	20	April–June
*Epinephelus akaara*	Actinopterygii	Threatened	15	25	April–June
*Anguilla japonica*	Actinopterygii	Threatened	10	3	November–December
*Penaeus japonicus*	Crustacea	Economic	10	20	December–March
*Acropora solitaryensis*	Anthozoa	Threatened	7	5	April–May

Note: PLD: pelagic larval duration, Depth: experimental living depth, Spawning window: the species’ spawning period in the Yellow and East China Seas. Species were classified into endangered and economic species according to the number of distribution sites. The scientific names were displayed in order of PLD length.

**Table 2 biology-12-00396-t002:** The occurrence of study species and the number of released particles in each MPA.

MPA	Area (km^2^)	*Acropora solitaryensis*	*Apostichopus japonicus*	*Epinephelus akaara*	*Epinephelus bruneus*	*Nemipterus virgatus*	*Scomberomorus niphonius*	*Argyrosomus japonicus*	*Larimichthys crocea*	*Evynnis cardinalis*	*Portunus Trituberculatus*	*Anguilla japonica*	*Argyrosomus argentatu*	*Sepiella maindroni*	*Penaeus japonicus*
JS-1	514.55								✔	✔	✔	✔	✔	✔	✔
JS-2	2472.6							✔	✔	✔	✔	✔	✔	✔	✔
JS-3	780								✔	✔	✔	✔	✔	✔	✔
JS-4	47.1								✔	✔	✔	✔	✔	✔	✔
JS-5	15.46								✔	✔	✔	✔	✔	✔	✔
SH-1	241.55									✔	✔	✔	✔	✔	✔
SH-2	423.2									✔	✔	✔	✔	✔	✔
ZJ-1	549		✔	✔	✔	✔	✔	✔	✔	✔	✔	✔	✔	✔	✔
ZJ-2	218.4		✔	✔	✔	✔	✔	✔	✔	✔	✔	✔	✔	✔	✔
ZJ-3	484.78					✔	✔	✔	✔	✔	✔	✔	✔	✔	✔
ZJ-4	44.19					✔	✔	✔	✔	✔	✔	✔	✔	✔	✔
ZJ-5	57		✔	✔	✔	✔	✔	✔	✔	✔	✔	✔	✔	✔	✔
ZJ-6	30.8					✔	✔	✔	✔	✔	✔	✔	✔	✔	✔
ZJ-7	306.69			✔	✔	✔	✔	✔	✔	✔	✔	✔	✔	✔	✔
ZJ-8	311.04			✔	✔	✔	✔	✔	✔	✔	✔	✔	✔	✔	✔
ZJ-9	201.06		✔	✔	✔	✔	✔	✔	✔	✔	✔	✔	✔	✔	✔
FJ-1	67.83			✔	✔	✔	✔	✔	✔	✔	✔	✔	✔	✔	✔
FJ-2	22.6			✔	✔	✔	✔	✔	✔	✔	✔	✔	✔	✔	✔
FJ-3	24.44			✔	✔	✔	✔	✔	✔	✔	✔	✔	✔	✔	✔
FJ-4	34.9	✔	✔	✔	✔	✔	✔	✔	✔	✔	✔	✔	✔	✔	✔
FJ-5	69.11	✔		✔	✔	✔	✔	✔	✔	✔	✔	✔	✔	✔	✔
Distribution points		2	5	11	11	14	14	15	19	21	21	21	21	21	21
Number of Released particles		2000	5000	11,000	11,000	14,000	14,000	15,000	19,000	21,000	21,000	21,000	21,000	21,000	21,000

## Data Availability

The data and biophysical model configuration are available from the corresponding author upon reasonable request (caoling@sjtu.edu.cn).

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
