# Peer review of "Larval Dispersal Modeling Reveals Low Connectivity among National Marine Protected Areas in the Yellow and East China Seas"

_biology, 2023, doi:10.3390/biology12030396_

Round 1

Reviewer 1 Report

This study simulates larval connectivity between marine protected areas in China's Yellow Sea and East China Sea. The strength of connectivity between MPAs is evaluated by organism type and season, which can be a very important insight in the management of MPAs and the establishment of new ones. I found this manuscript to be excellent. While some English language expressions and figures could be modified, and the volume of Discussion could be simplified, overall only minor modifications are needed.

Individual comments are as follows:

1) I felt that the volume of the discussion was too much. As for the content, I think it is perfectly fine, but it would be easier for the reader to read if it were more concise.

2) Line 23: It is listed as “roosting depth”, but I am not familiar with this expression. Is it possible to change it to another word?

3) Line 37: It is listed as “coast currents”, but I think “coastal currents” is a more appropriate phrase.

4) Section 2.1 describes the types of MPAs, are there any specific regulatory differences between SMPA and MNR? Also, what depths are covered by these MPAs?

5) Is it possible to show the location of major rivers in Figure 1?

6) Line 141: It is listed as “ArakawaC”. Please correct to “Arakawa C”.

7) Line 147: Is it possible to show the location of Yangtze Estuary in Fig. 1?

8) Line 163: You are running the simulation for the year 2016, what was the reason for selecting this year? It is possible that there can be significant differences in the physical field from year to year, but was 2016 an average physical field?

9) Table 1 shows the PLD and Depth values, but it would be easier to read if the numbers were right-aligned. Also, in the caption of the table, it is easier to understand if it is stated that the scientific names are shown in the order in which the PLDs are long.

10) Line 212: With respect to the 15% mortality rate for larvae, does it always give a constant mortality rate regardless of water temperature?

11) Line 274: Is it possible to show the location of the Yangtze River in Fig. 1?

12) Line 281: I could not find the location of Hangzhou Bay.

13) Line 282 and Fig. 2: Season 4 is listed as “Oct to Feb”, but is it not Oct to Dec. Any reason for making it Feb?

14) Figure 3: This may be a minor point, but the color model looks similar to Fig. 2. Is it possible to use different color models for depth and abundance?

15) Figure 5 and Figure S1-S4: The size of the text in the figures is too small.

16) Line 460: I am a little confused about the expression “fresh research”, is it possible to change “fresh” to another word?

17) Line 500: You wrote “good news”, but I don't think it is appropriate in a paper; how about using notable, remarkable, results, findings, etc.?

18) Line 524: It is listed as “seasons 2 and 4 (winter and summer)”, does season 2 correspond to winter?

19) Line 657: It is listed as “ocean progresses”, what does this mean?

Reviewer 2 Report

This manuscript is a really nice look into the connectivity of MPAs in real-time in an understudied system.  The novel use of a model as opposed to costly and time-intensive techniques is a really great addition to the literature. Thank you for that.

A few notes: 

Throughout the manuscript, the authors refer to the MPAs as being "not yet" connected.  This connotation implies that there is evidence that the MPAs will one day without doubt be connected and that the only variable is time.  I am not convinced you have the evidence to support that assumption. This statement occurs in the abstract (line 10) and elsewhere throughout the manuscript, including the conclusion.  I would caution the authors in this word choice--it is not representative of the data you have presented.  Instead the statements in the discussion are far more accurate: (line 419) Connectivity is very low.  No interpretation of what the future would hold for these MPAs. 

My second concern is about the use of network theory to determine connectivity in this system.  For example, you have for all intents and purposes described an ecological gradient across latitudes, not a connected network which requires reciprocal movement between nodes.  All of your movement is uni-directional as far as you have found.  You even agree to this in L394.  I think it makes your conclusions just observations.  I do not really think you can conclude anything from this study except for this method wasn't really great at giving us a definitive answer. 

Line edits: 

L89: dont use "hardly" it is imprecise and has various meanings.

Table 2:  Fix these headings--they are brutal.  The check marks are square root symbols--use the correct symbol.

L221 species not specie

L267-269: remove

L333 Penaeus japonicus needs to be italicized

Final comments: 

The discussion is *far* too long--almost 5 pages.  And, it is off-topic.  Focus on what you want to say rather than a complete review and policy report.  I think the discussion gets pretty far off-topic. 

Reviewer 3 Report

The paper is very interesting and addresses one of the crucial issues regarding the need to safeguard the marine environment from an ecological and economic point of view according to the principles of the blue economy. In fact, understanding how to capture the larvae of both endangered and commercially interesting species represents a global challenge.

The Authors have exhaustively explained the general problem, the utilized methods, the obtained results without neglecting details. They also well explained the possible limitations of their study and how the gaps can be filled.

The references are appropriate and refer to a very wide time interval, without neglecting either the oldest or the most recent ones. Nevertheless, in some cases they appear too redundant, so I suggest to reduce their number, maybe deleting the older ones.

Only a few suggestions are reported below.

Introduction:

Line 47: It should be better substitute the verb “constructed” with “established” being MPAs

Mat&Met:

Lines 114-122: I suggest deleting this list from the text and to add a table in Figure 1 with the names of the MPAs

Lines 144-147: It would be useful to add a picture reporting the computational grid utilized in the study

Table 2: The names of the species are too fragmented; please, rotate them 90 degrees to obtain unbroken words

Results:

Lines 267-269: Please, delete

Line 333: Make Penaeus japonicus in italics

Line 334: “Conversely, Acropora solitaryensis only occurred at two MPAs…”. Observing both Figure 4 and Figure S4 this assertion does not appear to be correct; please, verify the accuracy of the sentence

Discussion:

Line 503: It should be better substitute “construction” with “institution” being MPAs

References:

Reference 41 (Lines 789-790): add the year of publication (or “in press”)

Figures and Tables:

The captions are well explained so the reader can immediately understand a figure or a table. I suggest indicating the position of Yangtze River estuary on figures 2 and 3 to better follow the text.
